# OpenReview forum: "ContextBench: Modifying Contexts for Targeted Latent Activation and Behaviour Elicitation"
_ICLR.cc/2026/Conference — ICLR 2026 Poster_

### Official Review · Reviewer_NTHr · 2025-10-19

**Soundness:** 2
**Presentation:** 2
**Contribution:** 2
**Rating:** 2
**Confidence:** 4

**Summary:**

This paper investigates context modification, the generation of fluent inputs aimed at eliciting targeted model behaviors, and proposes a dataset for context modification ability evaluation. Furthermore, the authors propose two enhancements to a white-box context modification method, EPO, enabling its variants to achieve SOTA performance in both fluency and elicitation.

**Strengths:**

1. This paper incorporates several frontier techniques, including SAE and LLaDA.
2. In the proposed benchmark dataset, the authors include diverse scenarios and tasks to comprehensively evaluate the performance of context modification methods.

**Weaknesses:**

1. The text in Fig. 1 is hard to read. Since the figure is not complicated, I recommend using a larger font size.
2. I have reservations about the soundness of both contributions of this paper.

    i) For the proposed dataset, the number of samples seems too small to adequately represent the three tasks, especially given the additional subtasks under each. Moreover, since the dataset is divided by tasks, it is unclear whether these tasks comprehensively capture the concept of context modification, or whether they are representative enough for context modification. Additionally, using cross-entropy to measure text fluency may be somewhat outdated, despite its high correlation with human judgments. The authors may consider complementing it with learned metrics, which can really measure the deeper semantic appropriateness and contextual coherence.

    ii) For the two enhancement to EPO, first the authors should briefly introduce the core idea of EPO in the introduction. Second, without an explanation of why you chose EPO for improvement, these enhancements would seem random or incremental. Third, the implementation is relatively simple and heavily relies on GPT-4o and LLaDA. Will these variants still work well without such powerful auxiliary models? If not, why not directly build on these models instead?
3. The experimental comparison is not convincing. A GPT-4o-assisted variant is naturally more fluent than its baseline and more task-effective than GPT-4o itself. Since there are various prompt engineering methods, is it possible to adapt some SOTA approaches to the tasks introduced and compare the proposed methods with them?

**Questions:**

1. Could you briefly explain the mechanisms of honey-potting techniques mentioned in L38? I was wondering why you use this example right after stating your research focus.
2. Can I interpret context modification as a red-teaming process leveraging LLMs' prompt sensitivity, or as a subtask of prompt engineering? If so, what makes context modification stand out as a significant task?
3. Some terminology should be made consistent throughout the paper. For example, LLaDA vs. LLaDa.

---

> ### Author Response · Authors · 2025-11-24
>
> We thank the reviewer for their constructive feedback, and for recognising the “diverse scenarios and tasks which comprehensively evaluate the performance of context modification methods”. We address each concern below:
> ## Dataset
> **Size of the dataset**
>
> * SAE activation: We add 103 features from the LLaMA Scope 132K SAE release [2]  to 102 Gemma Scope features [1]  (total 205). These are hand-curated for diversity along density, vocabulary diversity and locality axes, giving broad coverage of the feature space.
> * Story Inpainting: We increased effective stories from 67 to 500 by writing 33 additional stories and giving each 5 target words, adding an additional task dimension.
> * Backdoors: 10 models across 4 tasks reflect a challenging frontier problem and sufficient variety to measure incremental progress.
>
> **Representativeness**
>
> We define context modification as generating fluent prompt edits that activate specified latents or elicit particular behaviours. We select tasks that decompose this into:
>
> 1. Core capabilities:
>     * SAE Activation tests the fundamental ability to maximise a wide range of latent features.
>     * Story Inpainting tests fluent in-context modification that change next-token predictions.
> 2. Hard-to-solve, real-world safety scenarios (backdoor trigger recovery) where white-box methods provide advantages over black-box.
> ## Fluency Metric
> Cross-entropy remains a standard proxy for fluency in recent work [3,4]. We expand our human study to 5 annotators, demonstrating that mean human fluency ratings correlate strongly with negative cross-entropy (Pearson r=0.94), confirming cross-entropy as a good proxy.
>
> We agree that complementary metrics are valuable. However, reference-based metrics (e.g. BLEURT, BERTScore [5,6]) are not generally applicable here because we do not assume reference contexts, and we want to allow any high-fluency, high-activation solution. Using LLM-as-a-judge is a valid alternative, but stifles objectivity and reproducibility.
> ## EPO and our modifications
> **Clarifying EPO.** We thank the reviewer for noting EPO needs clearer introduction; we address this in l.73-77.
>
> **Why EPO?** EPO is SoA for targeted latent activation. Section 4.2 now gives technical explanation of how EPO-Assist and EPO-Inpainting integrate fluency improvements while preserving gradient-based targeting.
>
> **Dependence on helper models.** In response to your feedback, we test sensitivity to helper model strength. We swapped GPT-4o with the lighter GPT-4o-mini for the Story Inpainting task. These experiments suggest that smaller models provide the fluency benefit of EPO-Assist but are slightly less adept at the local exploration for maximum activation. We add results to Appendix A.5.
>
> **Why not just use GPT-4o/LLaDA?** Black-box models cannot directly optimize SAE features; they have no access to model internals. Table 3 shows GPT-4o achieves inferior activation compared to EPO methods (97.3% win rate for EPO vs GPT-4o). Our modifications combine the complementary strengths: EPO's gradient-based targeting with LLM fluency.
> ## Baselines
> Combining gradient-based optimisation with LLM assistance requires careful design to not disrupt EPO's gradient signal. Empirical validation is also needed. That EPO-Assist empirically Pareto-dominates both vanilla EPO and GPT-4o alone (Fig. 3, 4, 5) is what the benchmark demonstrates.
> We compare against established methods appropriate to each task. Regarding other prompt optimisation methods (e.g. StablePrompt [7]), these approaches optimise for downstream task metrics, and lack white-box access. FLRT [8] is the most comparable work but requires a student-teacher setup (>> computationally expensive).
> ## Honey-potting techniques
> Honey-potting creates natural-looking inputs to reveal when a model behaves differently than intended, e.g. a deceptive model behaving well during evaluation but maliciously when it believes it is deployed. Context modification could generate inputs that avoid a model's "is this an evaluation?" latent, effectively “tricking” the model and exposing contexts for malicious behaviour.
> ## Context modification vs. red-teaming
> Context modification investigates dormant model capabilities by targeting specific latent features to elicit behaviors. Fluent inputs are more likely to occur in deployment and reveal generalisable patterns that trigger similar behaviours.
> Significance:
> * Interpretability: Recent research shows important behaviours (e.g., refusal) are mediated by linear directions in activation space [9]. Context modification identifies inputs that naturally activate these directions.
> * Safety evaluation: Modifying inputs to trigger specific latents tests whether models detect evaluation contexts, exhibit conditional behaviour, or hide capabilities.
> ## Clarity and Terminology
> We agree Fig. 1 was hard to read and increased the font size. We fix the inconsistencies in “LLaDA” and have done another pass of the manuscript.

---

> > ### Author Response · Authors · 2025-11-24
> >
> > ## References
> > [1] Gemma scope: Open sparse autoencoders everywhere all at once on gemma 2, Lieberum et al., 2024
> >
> > ⁠[2] Llama scope: Extracting millions of features from llama-3.1-8b with sparse autoencoders, He et al., 2024.
> >
> > [3] Automatically auditing large language models via discrete optimization, Jones et al., ICLR, 2023.
> >
> > [4] Autodan: interpretable gradient-based adversarial attacks on large language models, Zhu et al., NeurIPS, 2023.
> >
> > [5] BLEURT: Learning robust metrics for text generation, Sellam et al., ACL, 2020.
> >
> > [6] Bertscore: Evaluating text generation with bert, Zhang et al. ICLR, 2020.
> >
> > [7] StablePrompt: automatic prompt tuning using reinforcement learning for large language models, Kwon et al., 2024.
> >
> > [8] FLRT: Fluent Student-Teacher Redteaming, Thompson et al., 2024.
> >
> > [9] Refusal in Language Models Is Mediated by a Single Direction, Arditi et al., NeurIPS, 2024.

---

> > ### Comment · Reviewer_NTHr · 2025-11-27
> >
> > Thanks for the clarifications and the additional experimental results. I would like to maintain my score at this point.

---

> > > ### Author Response · Authors · 2025-11-27
> > >
> > > We thank the reviewer for their thoughtful feedback on our submission. We have addressed each point with evidence, clarifications, and/or additional experiments. If any concerns remain unresolved, we kindly ask the reviewer to specify which ones, explain why our responses are insufficient, and indicate what would address them. Below we summarise how each concern was addressed and pose specific questions to the reviewer.
> > >
> > > 1. **Dataset soundness**:
> > >
> > > Concerns were raised about dataset size and representativeness. We responded by:
> > >  * Expanding the dataset substantially (SAE features from 102 to 205; Story Inpainting from 67 to 500).
> > >  * Providing an explanation of how our tasks decompose the core capabilities required for context modification (latent activation, fluent in-context modification, trigger of hidden capabilities).
> > >
> > > **Please specify**: Which aspects of context modification does the reviewer believe our tasks fail to capture? What additional tasks would you consider more representative?
> > >
> > > 2. **Fluency metrics**:
> > >
> > > Concerns were raised about the use of cross-entropy. We responded by:
> > >  * Explaining why learned metrics are inapplicable when no reference contexts exist.
> > >  * Conducting expanded human evaluation showing strong correlation (Pearson r=0.94) between human judgments and cross-entropy.
> > >
> > > **Please specify**: Given our empirical validation of cross-entropy could the reviewer (a) explain what it is that cross-entropy is failing to achieve, and (b) what specific learned metric would the reviewer propose that doesn't require reference texts.
> > >
> > > 3. **Choice of EPO and baseline comparisons**:
> > >
> > > Concerns were raised about the choice of EPO and the lack of other method comparisons. We responded by:
> > >   * Clarifying that EPO is SoA as a white-box method for targeted latent activation.
> > >   * Clarifying that prompt-engineering methods without access to model internals cannot directly optimise latents and demonstrating this using GPT-4o as a fluent black-box method (Table 3: 97.3% win rate for EPO vs GPT-4o).
> > >   * Clarifying that our baselines were chosen to capture the key trade-offs in context modification:
> > >     * GCG (white-box) approximates an upper bound on elicitation strength without fluency constraints.
> > >     * GPT-4o (black-box) represents SoA fluent prompting without access to internals.
> > >     * EPO (white-box) as direct comparison.
> > >     * Human performance: indicates whether latents can be activated with human intuition alone, with ground truth fluency.
> > >
> > > **Please specify**: Given our principled selection of baselines, which specific SoA prompt-engineering method would the reviewer propose for comparison, and how would it optimise for arbitrary SAE features?
> > >
> > > 4. **EPO enhancements**:
> > > Concerns were raised about the missing explanations, the simplicity of our EPO enhancements, and the  reliance on helper models. We responded by:
> > >   * Providing extended technical explanation of our EPO enhancements EPO-Assist and EPO-Inpainting (now Section 4.2) – clarifying that integrating gradient-based targeting with LLM fluency is non-trivial.
> > >   * Clarifying that our variants being more fluent and more task-effective than GPT-4o is precisely what we are trying to achieve: improving upon the SOTA approach.
> > >   * Testing sensitivity to helper model strength (GPT-4o-mini experiments in Appendix A.5)
> > >
> > > **Please specify**: Given that (a) our enhancements make substantial improvements to “vanilla” EPO and (b) empirically Pareto-dominate both EPO and GPT-4o alone, what technical flaw does the reviewer identify in our approach?
> > >
> > > We have provided expanded datasets, additional experiments, empirical validations, and theoretical justifications. We respectfully ask the reviewer to provide specific technical or empirical reasons why our responses are inadequate so we can respond to them.

---

> > > > ### Comment · Reviewer_NTHr · 2025-11-28
> > > >
> > > > 1. I appreciate the authors' time and effort in expanding the size of the benchmark. However, my main concern lies in the **construct design** rather than the **volume**. For example,
> > > >
> > > >     i) **Overly broad task definition**. While the high-level objective of *"generating fluent prompt edits that activate specified latents or elicit particular behaviors"* is desirable, it is almost a definition of prompting itself. This makes it difficult to justify why the proposed tasks, rather than many other plausible ones including persona elicitation, emotion induction, and divergence control in open-ended QA, form a complete and principled taxonomy. Moreover, although the reference to latent feature activation gives the definition some specificity, there remains a conceptual gap between SAE features and observable behavioral patterns.
> > > >
> > > >     ii) **Weak shared construct**. For the three proposed tasks, I agree that they each touch on different aspects of CM, but they also have conceptual issues and differ so substantially in nature that they do not lie on the same conceptual axis.
> > > >     - *SAE activation* is mechanistic and tightly coupled to specific SAE models and source LMs, so the features are neither well-established nor shown to be generalizable, and the criterion of selecting *“interesting and diverse”* (L161-162) features from Gemma and Llama (L148-149) lacks rigor.
> > > >     - *Story inpainting* primarily targets natural language coherence and narrative completion. Its objective is semantic continuation rather than latent activation. It could be viewed as complementary to the first task (that is good!), but it would be more principled to *observe both phenomena within the same response* rather than *treating them as separate task categories*.
> > > >     - *Backdoor* is framed as trigger detection rather than actual defense (L186-187), which feels conceptually odd to me. Detecting an abnormal trigger is typically easier in a backdoored model than in a standard model, and such triggers should ideally be avoided in the first place.
> > > >
> > > >     A taxonomy is valid only when the tasks reflect the same dimension of capability. Even if the tasks overlap in their aim of *“causing a model to exhibit a target internal state,”* their differing operational goals weaken the construct validity.
> > > >
> > > > 2. Combining CE with human judgments makes more sense. Maybe we can consider using LLM-as-a-judge for fluency evaluation, especially if GPT-4o is treated as the SOTA reference for fluency in this paper.
> > > >
> > > > 3. The claim that EPO represents a state-of-the-art method for white-box CM seems somewhat overstated. Given the broad definition of CM above, I believe there likely exists a range of methods using steering vectors, activation-space constraints, or context-aware fine-tuning that are originally proposed for prompt engineering but could be readily adapted to CM. I apologize for not having enough time to identify the most similar and best-performing ones for you. Consequently, since I am not convinced that EPO is clearly dominating this task, I reserve my judgment on these two points (3 & 4).

---

> > > > > ### Author Response · Authors · 2025-12-02
> > > > >
> > > > > We thank the reviewer for engaging substantively with our work. We address each point below.
> > > > >
> > > > > ## Construct Design of Context Modification
> > > > >
> > > > > We clarify the distinction between context modification and prompting:
> > > > > * Context modification assumes a pre-existing context that is edited to achieve a specific latent activation or behavioural change - analogous to image inpainting versus generation. The constrained editing problem introduces different challenges (fluency preservation, minimal perturbation, targeted activation).
> > > > > * Latent targeting provides specificity. Unlike prompting for task performance, context modification optimises for internal model states (SAE features, probe directions) or precise token predictions. This white-box access distinguishes our setting from prompt engineering.
> > > > > * Safety motivation shapes task selection. We selected tasks relevant to safety auditing: understanding what activates interpretable features, fluent manipulation that changes predictions, and trigger recovery. While persona elicitation and emotion induction are interesting, they are less directly relevant to our target applications (honeypotting, detecting sandbagging, surfacing triggers).
> > > > >
> > > > >
> > > > > ## Task Heterogeneity and Construct Validity
> > > > >
> > > > > Our benchmark adopts a formative measurement approach (Bollen & Lennox, 1991; Diamantopoulos & Winklhofer, 2001). In psychometrics, reflective models assume a latent construct causes observed indicators (appropriate for traits like depression). Formative models assume indicators collectively define the construct - items need not correlate because they together comprise competence. Classic examples include socioeconomic status and professional licensing exams.
> > > > >
> > > > > ContextBench follows the formative approach: tasks collectively define the capability for safe model auditing via fluent elicitation, without requiring a single latent factor. Each task requires fluent text achieving a precise target; they vary in what is targeted but together constitute the capability we measure. Validity derives from domain coverage, not factorial unity. We respectfully suggest the reviewer's concern reflects assumptions from reflective measurement that do not apply to formative capability benchmarks.
> > > > >
> > > > > ## Individual Tasks
> > > > >
> > > > > * *SAE Activation*: We expanded to two SAE families (adding LlamaScope on Llama-3.1-8B) to test cross-architecture transfer. Features were selected along three principled axes - density, vocabulary diversity, locality - based on prior work (Bloom, 2024; Lee, 2024), covering all 27 combinations. The conceptual gap between SAE features and behaviour is precisely why context modification is valuable: EPO can reveal when a "numerical patterns" feature actually fires on "1" (Figure 5b).
> > > > >
> > > > > * *Story Inpainting*: This intentionally isolates fluency capability. A method achieving high SAE activation but unable to produce coherent edits (like GCG) is incomplete. Changing predictions via fluent edits is directly safety-relevant.
> > > > >
> > > > >
> > > > > * *Backdoors*: Trigger detection is a precursor to defence - auditors who cannot identify triggers cannot patch against them. We acknowledge methods largely fail at multi-token recovery; we include this as a challenging target, not a solved problem.
> > > > >
> > > > > ## Fluency Evaluation
> > > > >
> > > > > We appreciate the suggestion to use LLM-as-a-judge. We distinguish two use cases:
> > > > > * Post-hoc evaluation: LLM-as-a-judge could complement cross-entropy for final output assessment. We consider this a straightforward extension and will note it in future work.
> > > > > * Within-loop optimisation: EPO requires a differentiable, fast-to-compute fluency signal for gradient-based search. Cross-entropy satisfies both requirements; LLM-as-a-judge does not (it is non-differentiable and costly). Our human validation  confirms cross-entropy is an adequate proxy for the optimisation setting.
> > > > >
> > > > > ## State-of-the-Art Status of EPO
> > > > >
> > > > > EPO is state-of-the-art for gradient-based, targeted latent activation with fluency constraints in discrete token space. We are unaware of methods that use gradients for arbitrary SAE activation while balancing fluency via a differentiable objective. The alternatives mentioned address different problems: steering vectors modify forward passes (not inputs), activation-space constraints require inference intervention, and fine-tuning modifies weights. We seek to characterise fixed models via input discovery.
> > > > >
> > > > > We hope this clarifies our design choices and remain committed to addressing remaining concerns.

---

### Official Review · Reviewer_aHDC · 2025-10-28

**Soundness:** 3
**Presentation:** 2
**Contribution:** 3
**Rating:** 4
**Confidence:** 4

**Summary:**

This paper introduces ContextBench, a new benchmark designed to evaluate methods for context modification in language models, which involves generating linguistically fluent inputs that activate specific latent features or elicit targeted behaviors. This research is motivated by AI safety, seeking to identify contexts that trigger problematic model behaviors before deployment, such as sandbagging or bypassing refusal mechanisms. The authors formalize this approach and present ContextBench with three task categories: SAE Activation (targeting internal features), Story Inpainting (testing fluency in natural contexts), and Backdoors (recovering hidden triggers). To address the trade-off between elicitation strength and linguistic fluency, the authors develop two novel enhancements to Evolutionary Prompt Optimisation (EPO), demonstrating superior performance compared to vanilla EPO and black-box methods like GPT-4o. The overall goal is to advance white-box techniques for interpreting and controlling language model behavior through careful input design.

**Strengths:**

- This paper tackles a relevant problem in AI safety literature and present a benchmark that can be potentially useful for the community.
- The benchmark includes tasks representative of practical safety use cases, such as finding triggers for backdoored models.
- The benchmark follows a multi-objective evaluation approach for context modification.
- The authors provide an (anonymized) github repo with instructions and the results seem to be fully reproducible.
- The paper develops two novel variations of Evolutionary Prompt Optimisation that empirically Pareto dominate previous methods in balancing efficacy and fluency.

**Weaknesses:**

- There is a sort of "inconsistency" on the mentioned contributions: in the abstract, the authors start by focusing on investigating a class of methods (which one?), while in the introduction they clarify by stating they propose a new benchmark and "two state-of-the-art methods that empirically Pareto dominate previous methods on this task". Which task?
- The paper is not particularly easy to read in some parts; the flow of the paper could be significantly improved. A few more concrete examples, although the same thing applies for the actual flow ideas:
	1. Why is there a "Background" and "Related Work" section? It looks like they belong within the same section.
	2. There's a subsection (4.1) with no content on it, only on its subsubsections. Can't
 they just be subsections on their own?
 	3. SAE is never defined in the main body of the paper, only on the appendices, Figure 13 where the role of the LLM is defined. EPO is defined both in the abstract and main body, but the acronym is used in the main body of the paper before being defined on page 3 (line 109).
- One of the core contributions of this paper is the ContextBench benchmark, yet there is not discussion on relevant/related benchmarks that a future reader might want to look into as well. For example, how do the Story inpainting tasks relate to, or could be used on existing literature? On the same note, recent work on inpainting such as "Inpainting-Guided Policy Optimization for Diffusion Large Language Models" use mathematics datasets, could a paper like this benefit from using this benchmark as well?
- The paper proposes two proposed methods seem to rely quite heavily on EPO, however there is no description or details provided, or related work regarding this method. Simultaneously, there is very little detail provided on the two methods the authors propose. From reading the short descriptions provided, they seem like a trivial modification of EPO, but this perception might be incorrect and possibly arising from the lack of information provided. I believe three short paragraphs on two core contributions of a paper is not sufficient to fully understand such a core component of the research presented here.
- In my opinion, adding one or two additional methods to evaluate the two proposed methods against would be quite important. Relying solely on GCG, Human evaluations and GPT-4o as a baseline seems insufficient to me. A broad description of the participants/human evaluators would be desirable (age group, education level, etc.)
- The human evaluation was conducted using solely two annotators, which seems insufficient.
- [Suggestion] Increasing the font size on Figure 1 would improve readability
- [Suggestion] A table summarizing the results on the Backdoor task would be useful (i.e., not in the appendix, since it's a core result).

**Questions:**

- Why use cross-entropy as a fluency metric if, as the authors recognize, it is an imperfect approach? What alternatives should readers consider? Why not use perplexity as a model fluency metric?
- I am curious on the choice of GCG as a solution for the tasks in your benchmarks; I was aware of this method as a jailbreaking approach, designed to introduce "nonsensical" prefixes to prompts that would enable the generation of an unsafe output, or a non-refusal. Is there any research supporting the use of GCG for anything else other than jailbreaking?
- Is this an IRB-approved study (or equivalent, if the authors are not based in the U.S.)? If so, this information should be present in the paper

**Details Of Ethics Concerns:**

Two human annotators were used, yet there is no information regarding the two annotators. How were they chosen? Were they compensated? Are they unbiased to the research conducted in this paper?

---

> ### Author Response · Authors · 2025-11-24
>
> We thank the reviewer for their useful feedback, which improved our paper.
>
> **Clarification of the motivation and contribution**
>
> We are motivated by an AI safety case for discovering LLM prompts that trigger problematic model behaviours before deployment. To this end we define an **approach** called context modification: automatically generating fluent modifications to text within a language model prompt that cause a model to display undesirable behaviours (such as e.g.refusal [1]) via latents.
>
> We create a **benchmark** of **tasks** designed to assess the capabilities required of methods to perform context modification, and a task representative of a real safety use case.
>
> We then look into **methods** that perform context modification. We develop novel enhancements to SoA EPO: LLM-assistance and diffusion model inpainting. We highlight a novel contribution that we omitted - we are the first to use SAE latents for this purpose.
>
> **Paper structure**
>
> We improved paper flow:
> * We move “Background” into the methods section (now Sec. 4). We keep “Related Work” focused on positioning context modification.
> * We clarify in Sec. 4 that 4.1 describes the tasks and 4.2 the evaluation, and add short introductory sentences to each subsection.
> * We add a definition of SAEs to the related work, and define EPO and SAE acronymsi n the abstract and again at first use in the main text.
>
> **ContextBench vs. existing benchmarks**
>
> To our knowledge, ContextBench is the first benchmark for fluent latent activation and context modification. Existing benchmarks cover individual components but not their combination:
> * SAE evaluation benchmarks (e.g. SAEBench [4]) focus on feature quality and interpretability,  but don't test whether SAE features can guide feature visualisation.
> * Story completion benchmarks (e.g. ROCStories [5]) evaluate narrative coherence and generation quality, not changes to steer towards specific tokens.
> * Backdoor benchmarks (e.g. BackdoorLLM [6]) emphasise detection over trigger reconstruction.
>
> Regarding Inpainting-Guided Policy Optimization [IGPO], both papers leverage diffusion model inpainting, but with different goals. IGPO’s datasets evaluate whether hints improve RL training efficiency, whereas our EPO-Inpainting uses inpainting around high-activation tokens in order to improve fluency while preserving EPO’s activation objective. IGPO’s datasets do not provide latent or token-level targets required for ContextBench.
>
> **Providing technical context for EPO modifications**
>
> We expanded the original Sec. 5.2 to clarify the design choices behind EPO-Assist and EPO-Inpainting. We detail the timing of interventions, the selection of what tokens to preserve, and the choice of inpainting model. We believe these are principled and non-trivial modifications rather than superficial tweaks.
>
> **Human evaluation study**
>
> We expanded our study to n=5 annotators on the same 80 sentences. Inter-rater reliability for mean fluency is strong (ICC(2,k) = 0.95). Mean human fluency ratings correlate strongly with negative cross-entropy (Pearson r=0.94), indicating that cross-entropy is a good proxy for human-perceived fluency.
>
> We did not record demographic information to keep the study low-risk. The annotators were all adults and proficient English speakers.
>
> **Baselines and evaluation choices**
>
> Our baselines were chosen to capture the key trade-offs in context modification:
> 1. GCG (white-box, no fluency term) approximates an upper bound on elicitation strength without fluency constraints and is equivalent to EPO without the cross-entropy term.
> 2. GPT-4o (black-box) represents SoA fluent prompting without access to internals.
> 3. EPO is our white-box baseline and SoA context modification.
> 4. Human performance: provides ground truth for evaluating whether fluency metrics capture linguistic naturalness.
>
> We acknowledge that alternative methods exist for each separate subtask in our benchmark. However, ContextBench deliberately probes different aspects of context modification with its three distinct tasks.
>
> **Use of cross-entropy as a fluency metric**
>
> Cross-entropy is widely used as a reference-free fluency proxy in prompt-optimisation work [7,8] and integrates directly into EPO’s objective. Perplexity is simply the exponential of cross-entropy, so optimising one is equivalent to optimising the other.
> We also considered but chose not to use them as primary objectives:
> LLM-as-a-judge introduces model- and prompt-dependency and is costly
> Learned metrics like BLEURT [9] and BERTScore [10] require reference texts; however, we want to allow any fluent string that achieves the target activation.
>
> **Clarity and formatting**
>
> We rearrange, make the text bolder and larger in Figure 1 to improve readability. We move the table summarising the results on the Backdoor task to the main text.
>
> **Ethics approval**
>
> We confirm that the human study received approval from our funding institution and have expanded our ethics statement.

---

> > ### Author Response · Authors · 2025-11-24
> >
> > **References**
> >
> > [1] Refusal in Language Models Is Mediated by a Single Direction, Arditi et al., NeurIPS, 2024.
> >
> > [2] Fluent dreaming for language models, Thompson et al., 2024.
> >
> > [3] Universal and transferable adversarial attacks on aligned language models, Zou et al., 2023.
> >
> > [4] SAEBench: A Comprehensive Benchmark for Sparse Autoencoders in Language Model Interpretability, Karvonen et al., 2025
> >
> > [5] A Corpus and Evaluation Framework for Deeper Understanding of Commonsense Stories, Mostafazadeh et al., 2016.
> >
> > [6] BackdoorLLM: A Comprehensive Benchmark for Backdoor Attacks and Defenses on Large Language Models, Li et al., 2024.
> >
> > [7] Automatically auditing large language models via discrete optimization, Jones et al., ICML, 2023
> >
> > [8] Autodan: interpretable gradient-based adversarial attacks on large language models, Zhu et al., 2023
> >
> > [9] BLEURT: Learning robust metrics for text generation, Sellam et al., 2020,. ACL
> >
> > [10] Bertscore: Evaluating text generation with bert, Zhang et al., 2019

---

### Official Review · Reviewer_ccTx · 2025-11-03

**Soundness:** 3
**Presentation:** 3
**Contribution:** 4
**Rating:** 8
**Confidence:** 4

**Summary:**

The paper introduces a novel benchmark to assess the potential of linguistically fluent context modifications strategies to trigger targeted features in the latent space of LLMs. Subsequently, the authors introduce two enhancements to EPO to balance model elicitation and linguistic fluency.

**Strengths:**

- The authors created a novel, comprehensive, and practical safety benchmark for LLMs, a timely and valuable contribution. Backdoored models are particularly innovative.
- Clear paper providing a deeply quantitative and technical approach to safety evaluation
- Rich Appendix on data, model training and evaluation methods
- The LLM-improved EPO to increase fluency and in-painting strategy with diffusion language model
- Insightful read

**Weaknesses:**

- Valuable benchmark and innovative methodologies, but somewhat missing clear and actionable conclusions
- Already dense paper, and more details on the SAE itself would have been welcome in Appendix

**Questions:**

/

---

> ### Author Response · Authors · 2025-11-24
>
> We sincerely thank the reviewer for their positive feedback - especially the comment that it was an insightful read and that the benchmark is comprehensive and practical. We agree with the reviewer’s points and respond to them below.
>
> **Actionable conclusions**
>
> We note some actionable takeaways for different stakeholders:
> * For practitioners.
>     * We think these methods can be used for auto-interpretability of SAE features (and other dictionary learning techniques), as we find interesting cases where our methods highlight interesting examples that current Neuronpedia auto-interp does not highlight (e.g. Figure 5).
>     * EPO-Inpainting achieves the best balance between elicitation strength and fluency, making it our recommended choice between our modifications.
>     * We hope these methods can be used to probe for dangerous capabilities as part of model auditing processes.
>     * Unsurprisingly, white-box methods substantially outperform black-box approaches for internal feature targeting. We also show that SAE features can effectively guide feature visualisation in language models.
> * For future research.
>     * Multi-token trigger recovery remains fundamentally unsolved, requiring new sequence-aware methods or alternative objective formulations.
>     * We believe there is an interesting link between specification gaming strategies and jailbreak techniques, motivating adversarial robustness research.
> * For benchmark users.
>     * We hope our benchmark can be used by the community, and welcome any Github feature requests or commits - and particularly any improved methods.
>
> **SAE details**
>
> We provide details on the two SAE families that are used, which we add to the appendix:
>
> GEMMASCOPE-RES-16K [1]
> * Model: Gemma2-2B
> * JumpReLU SAE architecture
> * ~16,384 latents trained on post-MLP residual stream activations
> * 4B training tokens
> * Learning rate: 7e-5, Adam optimiser
> * Inputs normalised to unit mean squared norm
>
> LLAMASCOPE-RES-131K [2]
> * Model: Llama-3.1-8B
> * Top-K SAE architecture with JumpReLU post-processing
> * Trained on the post-MLP residual stream
> * SlimPajama corpus
> * Learning rate: 8e-4, Adam optimiser
>
> We also point out that the HuggingFace benchmark contains a table cataloging all SAE features with their Neuronpedia IDs, layer numbers, and categorisations for full reproducibility.
>
> **References**
>
> ⁠[1] Gemma scope: Open sparse autoencoders everywhere all at once on gemma 2, Lieberum et al., 2024
>
> ⁠[2] Llama scope: Extracting millions of features from llama-3.1-8b with sparse autoencoders, He et al., 2024.

---

### Official Review · Reviewer_ukNL · 2025-11-04

**Soundness:** 3
**Presentation:** 3
**Contribution:** 3
**Rating:** 6
**Confidence:** 3

**Summary:**

This paper introduces ContextBench, a benchmark for evaluating methods that generate linguistically fluent inputs to activate specific latent features or elicit targeted behaviors in language models, with potential applications to safety-relevant scenarios such as preventing jailbreaking and sandbagging. The benchmark comprises samples across three categories: (1) producing SAE Activation for selected semantic features; (2) Story Infilling to maximize a specific next word prediction for specific token pairs, and (3) Trigger detection for backdoored models. To improve current performances on these task, the authors propose two enhancements to the gradient-based Evolutionary Prompt Optimization (EPO) method: EPO-Assist, which periodically queries GPT-4o to generate fluent variations, and EPO-Inpainting, which uses the bidirectional diffusion model LLaDa to infill non-critical tokens while preserving high-activation tokens. Experiments demonstrate that these variants achieve better trade-offs between elicitation strength and fluency compared to vanilla EPO, GCG, and GPT-4o baselines. However, results on the backdoor detection tasks show only partial success, particularly struggling with multi-token triggers.

**Strengths:**

* Originality: ContextBench provides useful settings to evaluate the effectiveness of input synthesis methods across various settings that are novel and well-motivated for safety applications. The use of diffusion models for iterative editing represents a creative combination of existing methods to improve the EPO technique.
* Quality: The experimental methodology is generally sound with appropriate controls: multiple random initializations, statistical significance testing, human validation of the fluency metric, and comprehensive ablations across feature dimensions (density, vocabulary diversity, locality). The benchmark design shows careful consideration of task diversity and difficulty gradients.
* Clarity: The paper is provides clear motivation, explicit task descriptions, and good use of illustrative examples (e.g., Figure 4 showing how EPO discovers that a "celebrity" feature activates more on historical figures than the max-activating examples suggested). Implementation details are thoroughly documented in the appendices.
* Significance: Eliciting specific latent representations or predictions from language models is an underexplored area with promising real-world applications for detecting problematic model behaviors. The benchmark could enable systematic progress on this problem, with potential applications to security and interpretability research.

**Weaknesses:**

* While cross-entropy is validated against human judgments (ρ=0.92), the paper acknowledges it "promotes generic sentences, word repetitions, and creates dependencies on the specific LLM used.", as shown in the examples from Figure 4. The  threshold of 3-9 for cross-entropy also lacks principled justification. For the human evaluation part, only 80 examples were covered with just two annotators. A more robust fluency evaluation would strengthen claims, especially given that fluency is positioned as a core contribution. Authors should consider metrics like diversity (self-BLEU), semantic coherence (sentence embeddings), or larger-scale human studies to strengthen their claims.
* The paper documents multiple instances where methods exploit shortcuts rather than providing genuine insights: (a) inserting target words directly, (b) exploiting polysemy (medical vs. behavioral "rash"), (c) using conjunctions to reverse implications, (d) question/task switching in backdoors. While acknowledged, these issues fundamentally undermine the interpretability goal. The authors mention manual inspection and cross-entropy filtering as mitigation but provide no systematic solution or quantification of how often gaming occurs. This is a critical limitation for safety applications.
* Results show only 5.1% success rate for single-token password recovery (EPO) vs. 2.5% (GCG), with complete failure on multi-token sequences. For the auditing task, GPT-4o identified generated prefixes as outliers 100% of the time. These results suggest the proposed methods are not yet practical for real-world backdoor detection. The disconnect between token-by-token optimization and multi-token trigger sequences appears fundamental but is not adequately addressed. Moreover, the use of backdoored models from other papers is questionable, if these can be gamed by single-token triggers (e.g., "Ukraine" in line 451) that defeat in part the purpose of multi-token evaluation.
* In terms of clarity, useful results such as those presented in Figure 7 are often relegated to the appendix but directly referred from the main text, making the discussion harder to follow. The "Background" section also feels out of place in its current position, as it relates more to the methods presented in Section 5, rather than the presentation of tasks from Section 4. The examples of Figure 2 can be made generic to avoid forward references to the method, and the method can be introduced at the beginning of Section 5. In general, the writing can be refined and figure/table sizing optimized to warrant the inclusion of additional relevant contents from the appendix.

**Questions:**

Can you quantify how frequently specification gaming occurs across all tasks? What percentage of "successful" outputs actually provide interpretable insights vs. exploit shortcuts? This is crucial for assessing practical utility.

---

> ### Author Response · Authors · 2025-11-24
>
> We thank the reviewer for their helpful feedback, which has helped us improve the paper.
>
> **Cross-entropy as a fluency measure**
>
> We agree that robust fluency evaluation is essential. Following the suggestion, we increased human evaluation to n=5 on the same m=80 sentences. Inter-rater reliability fo mean fluency score is strong (ICC(2,k) = 0.95). Mean human fluency ratings are strongly associated with negative cross-entropy (Pearson r=0.94), indicating cross-entropy is a good proxy for human-perceived fluency. The 80 sentences include outputs from EPO, EPO-Assist, GCG, and GPT-4o and span the full 5-point fluency scale, which we believe is sufficient for validating the metric.
>
> The 3–9 range was calibrated on human-written answers in the Story Inpainting task (min=2.06, max=8.50) and manual inspection of model outputs; e.g. the lower bound of 3 reliably filters generic, repetitive sentences.
>
> **On alternative metrics**
>
> * Diversity: Diversity metrics such as self-BLEU could be incorporated into the EPO objective to encourage multiple distinct prompts that activate the same latent feature, which could aid interpretability. We view this as a promising extension.
> * Semantic coherence: For Story Inpainting task, one could augment fluency with an embedding-based local coherence score between the inpainted sentence and its neighbours. For SAE Activation there is no multi-sentence narrative, so penalising semantic drift is less meaningful.
>
> **Shortcuts and interpretability**
>
> We agree that specification gaming is an important concern but do not think it undermines the interpretability goal. Context modification explicitly aims to uncover effective “shortcuts” that elicit behaviours. If unintended yet fluent shortcuts (e.g. backdoors) exist, we want to surface them; if not, the model may resort to more mundane behaviours (e.g. using conjunctions). Indeed, specification gaming can be insightful:
> * If an SAE feature fires on the use of the word “happy”, rather than the concept of happy, then EPO is correctly revealing that the feature behaves like a lexical detector. This is not a failure of interpretability but a property of the SAE feature.
> * In the Backdoor task, “task switching” and question injection resemble real-world jailbreak and prompt-injection strategies. Finding these shortcuts gives us mechanistic insight that is useful for safety applications.
>
> We agree it was an oversight not to quantify how often such gaming occurs. We manually inspected a random sample of n=80 Story Inpainting responses from GCG, EPO and EPO-Assist.
> * Insertion of the target word was the most common failure mode. It occurred frequently for GCG and EPO (40%) and was rarer for EPO-Assist (<10%). This arises because the Story Inpainting task uses target logit rather than SAE feature activation as the elicitation goal.
> * Polysemy and reversing implications via conjunctions were rare: we identified only one polysemy case and one implication-reversal for EPO and GCG.
>
> To mitigate insertion of the target word, one could discard candidate solutions in the EPO population that contain the exact target word, or develop a more sophisticated objective function (though we note that target logit difference is standard in the mechanistic interpretability literature).
>
> **Password recovery**
>
> We agree that our backdoor results show this problem remains unsolved, and we do not claim that EPO-Assist succeeds in this real-world safety task. We include this task in the benchmark to provide a goal (early ImageNet accuracy was ~50%) and include our EPO results as a case study that exposes where current discrete optimisation methods succeed and fail.
> * Single-token recovery: The 5.1% single-token recovery rate is modest but non-trivial; especially given the extremely sparse conditions (exact password token in a longer context).
> * Multi-token triggers and the token-by-token objective: We agree that the failure on multi-token sequences reflects a disconnect between token-wise optimisation and sequence-level triggers. We already note this limitation in the paper (l. 421–424): gradient-based prompt optimisation struggles to coordinate long, precise trigger patterns. However, our experiments using probe-based latents as the objective (l. 465-469), which show  substantially better recovery, provide evidence that this challenge can be partly addressed with better latent features, motivating future work.
> * “Gaming” single-token triggers: The "Ukraine" example does not constitute gaming; it successfully creates the trigger (post-2023 headlines). However, it does reveal the backdoored models from Price et al. (2024) have simpler trigger structures than expected, which is itself a useful insight.
>
> **Clarity of presentation**
>
> We agree that the background section relates more closely to the methods in Section 5; we therefore move it to follow the benchmark tasks in Section 4 and use the additional page to bring key material from the appendix into the main body.

---

### Author Response · Authors · 2025-11-24
**Summary of revisions**

We sincerely thank all reviewers for their feedback, which we have used to improve our paper. We are encouraged by the positive reception, particularly Reviewer ccTx's "insightful read" comment and Reviewer ukNL's recognition that the benchmark "could enable systematic progress” on context modification.

We summarise our revisions in response to the feedback (these changes are highlighted in blue in the revised manuscript):
* **Substantially enlarged dataset:** SAE features expanded from 102 to 205 (adding LLaMA Scope); Story Inpainting grew from 67 to 500 effective examples.
* **Enhanced technical detail:** Section 5.2 now provides substantially more technical explanation of EPO-Assist and EPO-Inpainting design choices, and we added sensitivity analysis with GPT-4o-mini (Appendix A.5).
* **Quantified specification gaming:** Manual inspection of 80 Story Inpainting samples shows target-word insertion occurs in <10% for our EPO variant, with polysemy/conjunction gaming being rare.
* **Expanded human evaluation:** We increased our fluency validation study from 2 to 5 annotators across 80 examples, demonstrating strong inter-rater reliability (ICC=0.95) and correlation with cross-entropy (r=0.94).
* **Improved clarity and structure:** We moved the Background section to follow the benchmark tasks, added an EPO summary to the introduction, defined all acronyms properly, and enlarged Figure 1 for readability.

Multiple reviewers raised questions about EPO motivation, choice of fluency metric, and baseline comparisons. We have clarified that EPO is current state-of-the-art for targeted latent activation, validated cross-entropy thoroughly as a fluency proxy, justified our baseline selection, and explained why black-box methods fundamentally cannot maximally activate SAE features (97.3% EPO-Assist win rate vs. GPT-4o).

We acknowledge the backdoor task remains unsolved (providing a challenging target for future method development), and look forward to continuing this discussion during the rebuttal period.

---

### Meta-Review · Area_Chair_8SF4 · 2026-01-15

**Summary:**

Acceptance is recommended. The paper introduces ContextBench, a novel benchmark for generating fluent inputs to trigger specific latent features and behaviors, alongside improved Evolutionary Prompt Optimization (EPO) methods. The majority of reviewers (Scores: 8, 6, 4, 2) recognized the work's significance for AI safety and interpretability, praising the practical utility of the benchmark and the clear motivation.

**Reviewer Concerns:**

The authors provided a comprehensive rebuttal that addressed most critiques:

Dataset Size: The dataset was substantially expanded (SAE features doubled, Story Inpainting increased), addressing concerns about representativeness.

Fluency Metrics: The validity of cross-entropy as a fluency proxy was confirmed via an expanded human evaluation showing strong correlation ($r=0.94$), addressing concerns from multiple reviewers.Baselines: The authors clarified the necessity of white-box methods (EPO) over black-box prompting (GPT-4o) for targeting internal latents, showing significant win-rates.

**Reviewer Scores:**

The scores reflect a positive lean (8, 6, 4, 2). The single negative reviewer maintained their score based on a high-level disagreement regarding the "construct validity" of grouping these specific tasks, whereas other reviewers found the task diversity valuable for a safety benchmark. Given the extensive empirical improvements and the clear utility for red-teaming, the paper merits acceptance.

---

### Decision · Program_Chairs · 2026-01-26

Accept (Poster)